# Clinical status and cytokine profiles in patients with asthma or chronic obstructive pulmonary disease vaccinated against influenza

Mikhail Kostinov[1,2☯], Alexander Chuchalin[3☯], Anna Chebykina[4], Isabella Khrapunova[2], Alexander Cherdantsev[6‡], Irina Solov'eva[6‡], Nelli Akhmatova[1], Valentina Polishchuk[1], Nadezhda Kryukova[3], Aristitsa Kostinova[2]*, Anna Vlasenko[5], Marina Loktionova[2], Yvette Albahansa[2], Anna Shmit'ko[1], Lyudmila Shogenova[3]

1 I.I. Mechnikov Research Institute of Vaccines and Sera, Moscow, Russian Federation, 2 I.M. Sechenov First Moscow State Medical University, Moscow, Russian Federation, 3 Pirogov Russian National Research Medical University, Moscow, Russian Federation, 4 Kirov State Medical University, Kirov, Russian Federation, 5 Branch Campus of the Russian Medical Academy of Continuous Professional Education, Novokuznetsk State Institute for Advanced Training of Physicians, Novokuznetsk, Russian Federation, 6 Ulyanovsk State University, Ulyanovsk, Russian Federation

☯ These authors contributed equally to this work.
‡ AC and IS also contributed equally to this work.
* aristica_kostino@mail.ru

**Data Availability Statement:** All relevant data are within the manuscript and its Supporting Information files.

## Abstract

### Background

Influenza vaccine is a tool for preventing infection and reducing exacerbations in patients with asthma and chronic obstructive pulmonary disease (COPD). However, the associations between clinical outcomes and changes in the levels of inflammation markers have not been fully delineated. The purpose of this study was to investigate the clinical course and the changes in the levels of inflammation markers in patients with asthma or chronic obstructive pulmonary disease for one year after vaccination against influenza.

### Methods

The prospective study for one year included 34 patients with asthma, 20 patients with COPD vaccinated against influenza, both groups being under a basic maintenance therapy, and 26 healthy individuals vaccinated with the trivalent polymer-subunit (adjuvanted) vaccine, containing 5 μg of influenza virus strains and 500 μg of azoximer bromide. The levels of C-reactive protein (CRP) and serum cytokines (IL-2, IL-6, IL-10, and IL-17) were measured by enzyme-linked immunosorbent assay (ELISA) at baseline and 6 and 12 months after vaccination.

### Results

Over a year after vaccination against influenza, the frequency and duration of bronchopulmonary exacerbations significantly decreased both in patients with asthma and those with

**Funding:** The author(s) received no specific funding for this work.

**Competing interests:** The authors have declared that no competing interests exist.

COPD: by 1.9–2 and 2.2–2.5 times, respectively. There was also a significant reduction in the frequency and duration of hospitalization (by 2.0–2.5 and 2.3–3 times, respectively). Other changes observed over the one-year follow-up period included a 1.6-fold reduction (<0.01) in the need for outpatient care and a reduction in the number of courses of systemic corticosteroids (by 16.7%; <0.05) in asthma patients; and a 3.6-fold decrease (<0.05) in the number of courses of antibiotics in COPD patients. Twelve months after vaccination against influenza, the study participants had significantly lower IL-6 levels, and COPD patients, additionally, showed a reduction in IL-10 levels compared to baseline. Our study identified certain correlations between positive clinical outcomes of vaccination and levels of inflammation markers.

## Discussion

Analysis of the immunological, clinical and functional parameters in asthma and COPD patients showed that vaccination not only reduces the risk of influenza and other respiratory infections due to activation of non-specific protection, but also improves the clinical course of asthma and COPD.

## Introduction

Influenza referring to Orthomyxoviridae (specifically Influenza A and B viruses) and other respiratory infections are the leaders among the exacerbations of bronchopulmonary diseases [1–3]. There is no doubt that exacerbations in patients with chronic obstructive pulmonary disease (COPD) are combined with an increase in inflammation activity, therefore prolonging the post-exacerbation recovery period [4, 5]. In patients with frequent exacerbations, the quality of life (QoL) is worse, the reduction in pulmonary function is faster, and mortality rates are higher than in patients with less frequent exacerbations [6, 7].

Patients with COPD showed statistically significant changes in forced expiratory volume in the first second (FEV1) during the acute respiratory infections (ARIs), while influenza was associated with a greater decrease in FEV1 than other ARIs [8]. A significant deterioration in FEV1 was noted among patients with laboratory-confirmed influenza, in contrast to patients without laboratory confirmation of infection (48% vs. 24%). Each infection leads to the further deterioration of pulmonary function [9, 10]. The most formidable signs of severe influenza are the rapid progression to acute respiratory failure and the development of multi-lobe lung damage. Markers of bad prognosis are: reduction in saturation O2 (SatO2) less than 90%, tachypnea above 25 breaths/min, hemoptysis, hypotension, diarrhea, laboratory tests—thrombocytopenia, increased lactate dehydrogenase, creatine phosphokinase and creatinine [11, 12].

Every fourth patient with asthma after ARIs experiences a deterioration of pulmonary function, symptoms of bronchial obstruction and poor asthma control for 4–6 weeks [13, 14]. Influenza can often be the cause of hospitalizations. To date, vaccination against influenza, which has been used for decades, has proven its efficacy in reducing the frequency of exacerbations and the severity of COPD, asthma, and other comorbidities [15, 16].

Vaccination has a double effect: specific and non-specific. The specific mechanism includes the activation of humoral and cellular protection factors against the antigens that make up the vaccine composition [17–19]. The non-specific mechanism includes the stimulation of the

phagocytic activity of macrophages and neutrophils, as well as the secretion of lactoferrin, lysozyme, and interferon, which increases protective potential of the respiratory system regardless of the antigenic characteristics of microorganisms. That is, vaccination reduces the incidence not only of influenza but also of the other ARIs, both among adults and children by 25–65% [20, 21].

However, despite the discovery of many positive clinical and immunological effects of the influenza vaccination, investigations of immunization effect on the cytokine profile, C-reactive protein (CRP), which characterize the inflammatory process in patients with diseases of the bronchopulmonary system, as well as the relationship with clinical and functional parameters in patients with asthma and COPD vaccinated against influenza are currently limited. Studying these parameters will expand the concept of not only direct preventive, but also indirect therapeutic efficacy of the influenza vaccination. Purpose of this study is to investigate dynamics of CRP, serum cytokines (IL-2, IL-6, IL-10, IL-17) in patients with asthma and COPD, as well as to perform a correlation analysis with the clinical manifestations of the diseases within a year after vaccination.

## Materials and methods

### Materials

**Study design.** Primary objectives of the study were to assess baseline levels of CRP, serum cytokines—interleukins (IL) (IL-2, IL-6, IL-10, IL-17) during a period of remission in patients with asthma and COPD, and to study a history of the clinical course of the diseases for the previous 12 months. Secondary objectives were to analyze influenza vaccine's effect on the dynamics of the studied immune parameters (after 6–12 months) and the clinical effects of influenza immunization on the course of the underlying disease in patients with bronchial obstructive syndrome for 12 months.

The total number of individuals participating in the study was 82 patients, including 36 patients with asthma, 20 patients with COPD and 26 healthy subjects vaccinated against influenza.

Stage IV (postmarketing): an open-label, non-randomized, comparative, controlled study of observers was conducted in three centers in Russia. Patients were enrolled in the study group at the Research Institute of Pulmonology of the FMBA of Russia (Moscow), Pulmonology Department of Kirov Regional Clinic Hospital–Kirov Regional State Budgetary Healthcare Institution (Kirov) and I.I. Mechnikov Research Institute of Vaccines and Sera (Moscow). All patients were monitored by a pulmonologist and received treatment in accordance with the regulated protocols of clinical guidelines approved in the Russian Federation.

After preliminary studies and examination by a physician, taking into account indications and contraindications according to the instructions for the vaccine usage, as well as the official recommendations for immunization in the Russian Federation, informed consent was obtained from a participating patient, and the patient was sent to the immunization room. Vaccination was carried out by a nurse in compliance with aseptic and antiseptic measures, as well as the rules for the drug administration. After the vaccination, the patient was observed in a medical institution for 45 minutes to exclude the development of immediate reactions to the vaccine. A physician in charge asked the vaccinated person daily about his/her health and filled out the list of possible adverse reactions for the next 7 days. Within the next 12 months investigator—pulmonologist in each center, followed up all vaccinated patients and clinically assessed asthma and COPD course, organized and reminded for patients about repeated blood sampling and immunological studies in 6 and 12 months and transported to the laboratory specimens. All information about the vaccination, examination and the study data were recorded in

the official, standard, individual medical documentation of the patient, which can be used by an attending physician and healthcare organizers to monitor the performed studies.

**Evaluation of clinical parameters before and after vaccination.** After 6 and 12 months, the patients underwent a clinical and physical examination, including the collection of complaints and medical history, examination, frequency and duration of exacerbations of the underlying disease, courses of antibacterial chemotherapy and systemic corticosteroids, frequency and duration of hospitalization, days of work incapacity, frequency of outpatient visits during the year before and 12 months after the vaccination.

**Legal and ethical aspects of the study.** The study was conducted following the Declaration of Helsinki, the Guidelines of the International Council on Harmonization for Good Clinical Practice, and Russian regulatory requirements.

Vaccination of patients with asthma and COPD was carried out under the National Calendar of Prophylactic Vaccinations in the Russian Federation—a regulatory legal act, establishing the timing and procedure for conducting preventive vaccinations for citizens (Federal Law No. 157-FZ of September 17, 1998); Article 20 "Informed Voluntary Consent to Medical Intervention and Refusal of Medical Intervention" (Federal Law No. 323-FZ of November 1, 2011 "On Basics of Health Protection of the Citizens in the Russian Federation" (revised on April 3, 2017); Methodical Guidelines (MG) 3.3.1.1123–02 "Monitoring of Post-Vaccination Complications and their Prevention" approved by Chief State Sanitary Inspector of the Russian Federation on May 26, 2002).

The study protocol was approved by the local Ethics Committee of the Federal State Budgetary Scientific Institution I. Mechnikov Research Institute of Vaccines. Written informed assent was obtained from the patients before their enrolment in the study. All research was performed in accordance with relevant guidelines/regulations.

**Inclusion criteria.** Men and women over 18 years of age; patients with moderate or severe asthma or COPD, receiving bronchodilator and anti-inflammatory therapy with no signs of exacerbation at the time of inclusion in the study; informed consent signed by the patient.

**Exclusion criteria.** Acute infectious diseases, including tuberculosis; active phase of chronic viral hepatitis; mental disorders; renal or hepatic impairment; hypersensitivity to vaccine components; severe complications to previous vaccinations; pregnancy; inability of the patient to understand the essence of the study or give consent to participate in it.

**Study groups.** Group I: Asthma patients vaccinated against influenza (n = 34); Group II (n = 20): COPD patients; Group III (n = 26): healthy vaccinated subjects (as a reference group to assess changes in immunological parameters after vaccine administration).

The asthma diagnosis, its severity and control were confirmed by the history, clinical presentation and functional diagnostics (GINA).

The diagnosis of COPD was confirmed by the medical history, clinical presentation, X-ray, and functional diagnostics (GOLD). According to the GOLD criteria (FEV1/FVC <70% and 30% ≤FEV1 <80% of the reference value), the inclusion criteria are the following: patients with moderate to severe COPD aged >45 years, with a smoking history of >10 packs/year.

Patients were enrolled in the study after a relieved exacerbation, which occurred during the inpatient treatment in the pulmonary department of the Regional Clinical Hospital or were followed up on an outpatient basis. Patients with asthma and COPD were vaccinated within the period of 2–4 weeks of remission, against the background of maintenance basic therapy.

The characteristics of patients with asthma /COPD and healthy subjects included in the study are presented in Table 1. The groups of patients with asthma and COPD were comparable in terms of age and lung function parameters The groups of placebo and patients with respiratory tract diseases differed by sex, but no gender differences have not been revealed in

**Table 1. Characteristics of patients with asthma /COPD and healthy subjects included in the study.**

| Characteristics | Asthma group | COPD group | Healthy |
| --- | --- | --- | --- |
| | N = 34 | N = 20 | N = 26 |
| Males, abs./% | 14 / 39% | 19 / 95% | 8 / 31% |
| Females, abs./% | 22 / 61% | 1 / 5% | 18 / 69% |
| Age, years | 51.05 ± 1.73 | 58.5 ± 1.37 | 39.27 ± 3.25 |
| Disease history, years | 18.36 ± 2 | 15.7 ± 1.83 | - |
| Moderate, abs./% | 29 / 81% | 13 / 65% | - |
| Severe, abs./% | 7 / 19% | 7 / 35% | - |
| Steroid-dependent patients, abs./% | 6 / 17% | - | - |
| FEV1, abs. | 1.71 ± 0.13 | 1.08 ± 0.12 | - |
| FEV1, % due | 56.71 ± 3.57 | 32.25 ± 3.24 | - |
| FVC, abs. | 2.14 ± 0.19 | 1.48 ± 0.15 | - |
| FVC, % due | 55.70 ± 3.52 | 35.06 ± 3.29 | - |
| FEV1/FVC, % due | 68.33 ± 4.49 | 50.78 ± 3.49 | - |
| FEV1/FVC, % due (post-bronchodilator) | 83.38 ± 6.97 | 51 ± 2.59 | - |
| O2 saturation, % | 96.66 ± 0.25 | 95.62 ± 0.76 | - |
| 6-MX test, m | 378.94 ± 14.58 | 351.44 ± 14.53 | - |

Note: Continuous variables are presented as Mean ± SD, categorical variables are presented as number / percentage (%)

the formation of the immune response previously. All participants of the study were of average age of adult population aged from 36 to 60 years.

During the 1-year follow-up period after the vaccination, one patient with severe COPD withdrew from the study due to death caused by decompensated chronic respiratory and heart failure.

**Vaccine.** The trivalent polymer-subunit vaccine (NPO Petrovax Pharm LLC, Russia), was used for vaccination. The immunized study participants received a 0.5 mL dose of the vaccine during the period from the 20th of December, 2018 till the 18th of February, 2019. The vaccine includes protective antigens isolated from virus-containing allantoic fluid of chicken embryos, which are linked with immunoadjuvant azoximer bromide (Polyoxidonium). The vaccine contains 5 μg of hemagglutinin, epidemically relevant strains of influenza virus subtypes A (H1N1, AH3N2) and type B (total 15 μg of HA), and 500 μg of azoximer bromide. The vaccine contained viral strains recommended by the WHO for the 2018–2019 northern hemisphere influenza season for trivalent vaccines: A/Michigan/45/2015 (H1N1)pdm09-like virus, A/Singapore/INFIMH-16-0019/2016 A(H3N2)-like virus, B/Colorado/06/2017-like (Victoria lineage) virus. Pharmacological action of the vaccine: anti-influenza, immunomodulatory. The vaccine forms a high-level specific immunity against the influenza strains mentioned above. The protective effect after vaccination usually occurs in 8–12 days and lasts up to 12 months. Azoximer bromide, included in the vaccine, provides an increase in immunogenicity and stability of antigens, allows to increase immunological memory and significantly reduces the vaccination dose of antigens. Polyoxidonium as an adjuvant has proven itself in various in vivo and in vitro studies, especially among patients with immunosuppressive conditions [22–25].

## Methods

**Assessment of vaccine tolerance in patients with asthma and COPD.** According to the study protocol, all local and systemic post-vaccination reactions were recorded within 7 days after the immunization.

**Spirometry.** The spirometry study was performed using the Spiroanalyzer ST-95 following the recommendations of the European Respiratory Society at the baseline and 12 months after the vaccination. The main evaluated parameters were forced expiratory volume in one second ($FEV_1$), forced vital vessel (FVC), ratio ($FEV_1/FVC$), vital vessel (VC). The studied values were expressed as a percentage of the due, which simplifies the comparison of different groups of patients, eliminating the need for standardization by age, sex, weight, or height.

**Immunological method.** To study the levels of CRP and cytokines, donor blood was taken into disposable plastic vacuum sterile tubes with a volume of a minimum of 10 mL before the vaccination, and after 6 and 12 months after the vaccination. The tubes with blood were centrifuged at 3,000 rpm. The obtained blood serum samples from donors were frozen at -25°C.

Assay of the CRP level was performed by ELISA using the CRP–ELISA–Best (high sensitivity) reagent kit, Russia. Determination of IL-2, IL-6, IL-10, and IL-17 was performed by ELISA using Vector Best reagent kits, Russia, following the attached instructions for the generally accepted procedure.

The work was performed using licensed equipment of the Shared Use Center of the Federal State Budgetary Scientific Institution I. Mechnikov Research Institute of Vaccines and Sera.

**Statistics.** Descriptive statistics were used to process the numerical data. All numerical data are presented as Mean ± SD. To compare the data before and after the vaccination, the paired Wilcoxon test was used. To compare data at three time points (baseline, 6 months post-vaccination, and 12 months), the Friedman test with Dunn's multiple comparison was used for quantitative variables. For the binary variable, a generalized linear mixed model was used (mixed effects logistic regression). The differences were considered statistically significant at p <0.05. The Benjamini–Hochberg procedure was used (Tables 2 and 3) to control for the false discovery rate at a 5% level.

**Table 2. Analysis of medical care requests in patients with asthma and COPD against influenza.**

| Parameter | Before vaccination[1] | After vaccination | P value[2] | q value[3] |
|---|---|---|---|---|
| **Asthma group (N = 34)** | | | | |
| Frequency of BF exacerbations, episodes/year | 3.1 ± 0.2 | 1.6 ± 0.2 | **<0.001** | **<0.001** |
| Duration of exacerbations, days | 42.7 ± 3.3 | 19.6 ± 2.5 | **<0.001** | **<0.001** |
| Frequency of outpatient visits, cases/year | 14.0 ± 1.1 | 8.9 ± 0.9 | **0.002** | **0.003** |
| Frequency of hospitalizations, episodes/year | 2.5 ± 0.25 | 1.2 ± 0.2 | **<0.001** | **<0.001** |
| Duration of hospitalization, days | 36.4 ± 3.5 | 16.0 ± 2.7 | **<0.001** | **<0.001** |
| Number of courses of antibiotics per year | 1.4 ± 0.2 | 1.3 ± 0.2 | 0.92 | 0.92 |
| Number of systemic corticosteroids courses during the year | 2.1 ± 0.3 | 1.5 ± 0.2 | **0.04** | **0.046** |
| **COPD group (N = 20)** | | | | |
| Frequency of BF exacerbations, episodes/year | 2.9 ± 0.3 | 1.5 ± 0.2 | **0.01** | **0.017** |
| Duration of exacerbations, days | 45.4 ± 4.5 | 17.5 ± 3.4 | **0.004** | **0.009** |
| Frequency of outpatient visits, cases/year | 11.1 ± 1.4 | 8.9 ± 1.5 | 0.13 | 0.13 |
| Frequency of hospitalizations, episodes/year | 2.8 ± 0.3 | 1.1 ± 0.3 | **0.001** | **0.007** |
| Duration of hospitalization, days | 44.8 ± 4.9 | 15.1 ± 3.6 | **0.002** | **0.007** |
| Number of courses of antibiotics per year | 2.8 ± 0.6 | 0.8 ± 0.2 | **0.03** | **0.042** |
| Number of systemic corticosteroids courses during the year | 1.3 ± 0.3 | 0.8 ± 0.2 | 0.09 | 0.11 |

Note: Continuous variables are presented as Mean ± SD, categorical variables are presented as number / percentage (%)

[1] observation was 12 months before and 12 months after vaccination

[2] the paired Wilcoxon's test was used

[3] the Benjamini–Hochberg procedure was used (for the asthma group and COPD group separately)

**Table 3. Pre-bronchodilation parameters of spirometry in patients with asthma and COPD vaccinated against influenza.**

| Parameter | Baseline | After 12 months | P value[1] | q value[2] |
|---|---|---|---|---|
| **Asthma group (N = 34)** | | | | |
| VC, abc | 2.55 ± 0.18 | 2.84 ± 0.17 | 0.09 | 0.19 |
| VC, % | 63.21 ± 3.91 | 70.28 ± 2.57 | 0.06 | 0.19 |
| FVC, abc | 2.15 ± 0.24 | 2.49 ± 0.23 | 0.19 | 0.26 |
| FVC, % | 52.37 ± 3.91 | 58.84 ± 3.67 | 0.11 | 0.19 |
| FEV1, abc | 1.71 ± 0.17 | 1.86 ± 0.16 | 0.26 | 0.30 |
| FEV1, % | 52 ± 3.65 | 56.2 ± 3.46 | 0.08 | 0.19 |
| FEV1/FVC | 68.34 ± 4.49 | 64.59 ± 2.75 | 0.54 | 0.54 |
| **COPD group (N = 20)** | | | | |
| VC, abc | 2.08 ± 0.14 | 2.45 ± 0.15 | **0.01** | **0.049** |
| VC, % | 50.24 ± 2.75 | 60.06 ± 2.75 | 0.04 | 0.13 |
| FVC, abc | 1.62 ± 0.18 | 1.71 ± 0.13 | 0.28 | 0.38 |
| FVC, % | 37.69 ± 3.61 | 42 ± 2.74 | 0.07 | 0.12 |
| FEV1, abc | 1.13 ± 0.13 | 1.17 ± 0.08 | 0.49 | 0.49 |
| FEV1, % | 33.65 ± 3.43 | 35.76 ± 2.16 | 0.33 | 0.38 |
| FEV1/FVC | 52.7 ± 3.42 | 46.46 ± 3.2 | 0.06 | 0.12 |

Note: Continuous variables are presented as Mean ± SD

[1] the paired Wilcoxon's test was used

[2] the Benjamini–Hochberg procedure was used (for the asthma group and COPD group separately)

Statistical processing of the results was performed using GraphPad Prism (v.9.3.0 license GPS-1963924). All available data on the database were used to maximize the power and generalizability of the results [26].

## Results

### Post-vaccination period

The administration of the influenza vaccine in patients with asthma and COPD was not accompanied by the development of serious adverse events, and the incidence of moderate local reactions in the form of pain, redness, and induration at the injection site within 3 days after vaccination was observed only in 3 (8.3%) patients with asthma that do not require therapeutic measures. Systemic reactions were recorded in 4 (11.1%) vaccinated patients with asthma and in one patient with COPD (1.72%), which were characterized by the appearance of catarrhal events from the upper respiratory tract, but it was impossible to exclude episodes of intercurrent diseases within 7 days after vaccination. No patient needed to intensify the basic therapy. In healthy subjects, the post-vaccination period proceeded without any unusual events.

The follow-up of patients with asthma and COPD for a year after the vaccination revealed a significant reduction in the frequency and duration of exacerbations of the bronchial obstructive syndrome (BF) as well as the frequency and duration of hospitalizations (Table 2). In addition, patients with asthma showed a decrease in the need for outpatient care, a decrease in the frequency of prescription of systemic corticosteroids courses for 1 year; the number of patients who required course prescription of systemic corticosteroids during exacerbations decreased by 16.7%. In patients with COPD, in contrast to the observed patients with asthma, a decrease in the need for antibiotics was established as compared to the previous year before the vaccination.

**Table 4. CRP dynamics in asthma and COPD patients and healthy volunteers before and after vaccination against influenza (reference value: Up to 8 mg/L).**

| Group | n | Quant./ % [1] | Baseline | After 6 months | After 12 months | p |
|---|---|---|---|---|---|---|
| | | | M ± m | M ± m | M ± m | |
| COPD | 16 | Quant. | 22.44 ± 7.33# | 31.4 ± 11.95# | 17.71 ± 3.38# | 0.87 |
| | | % | 16 (100%) | 13(81%) | 13(81%) | 0.13 |
| Asthma | 32 | Quant. | 10.79 ± 1.88# | 9.27 ± 2.04# | 9.6 ± 1.94# | 0.64 |
| | | % | 26(81%) | 23(72%) | 19(59%) | 0.06 |
| Healthy | 18 | Quant. | 4.89 ± 1.29 | 4.61 ± 1.58 | 5.79 ± 2.17 | 0.37 |
| | | % | 0 | 0 | 0 | - |

Note

[1] The proportion of patients with laboratory parameters upper the reference values

#The parameter exceeds the upper limit of normal (8 mg/mL)

p: the significance of differences between the three evaluations in each group. For quantitative variables, the Friedman test was used, for binary variable, a generalized linear mixed model was used (mixed effects logistic regression).

**Spirometry parameters in vaccinated subjects.** Analysis of spirometry parameters in BA and COPD patients before vaccination and after 12 months of observation is presented in Table 3. In vaccinated BA patients after 12 months, there is a tendency to an increase in both volumetric and velocity parameters compared to the baseline. In COPD patients, an increase in VC(ab) was detected within the specified period compared to baseline. It should be noted that, despite the increase in VC, the indicator remains below standard values.

**CRP level in vaccinated subjects.** The level of CRP exceeded the conditional norm (up to 8 mg/L) in all study periods in asthma and COPD patients. (Table 4, Fig 1). In healthy subjects, the CRP was within the reference values. In the asthma group, the decrease in the proportion of patients with CRP levels above the reference level was observed at the trend level (p = 0.06).

**IL-2, IL-6, IL-10, and IL-17 levels in vaccinated people.** The level of IL-2 in COPD and BA patients and healthy volunteers during the entire follow-up period (12 months) was within the normal range (less than 10 pg/mL) (Table 5, Fig 2). In all groups, there was a tendency to a decrease in the level of IL-2 after 6 and 12 months (p >0.05), especially in COPD patients. With a larger number of observations, the differences might be more significant (p = 0.08).

The baseline IL-6 level in COPD patients was above normal (more than 10 pg/mL) and amounted to 12.13 ± 5.69 pg/mL. In BA patients and healthy volunteers, the IL-6 level was within the normal range (Table 6, Fig 2). After 6 and 12 months after vaccination against influenza, the IL-6 level in all groups was within the normal range, but its significant reduction compared to the baseline values was observed after 12 months.

In the COPD group, there was a statistically significant decrease in the proportion of patients with IL-6 levels above the reference level both 6 and 12 months after vaccination. In the asthma group, there was a statistically significant decrease in the proportion of patients with IL-6 levels above the reference level 12 months after vaccination.

The IL-17 level in all groups before vaccination, as well as after 6 and 12 months, was within the reference values (less than 5 pg/mL), although it was higher in BA patients than in COPD patients and healthy volunteers (Table 7, Fig 3).

IL-10 level in all groups at the time of vaccination, as well as after 6 and 12 months, was below the permissible values (less than 20 pg/mL) (Table 8, Fig 3). During the first 6 months after vaccination, there were no statistically significant changes in the IL-10 concentration in any group. Over the next 6–12 months, there was a significant reduction in the level of IL-10 compared to baseline in the group of patients with COPD and healthy people.

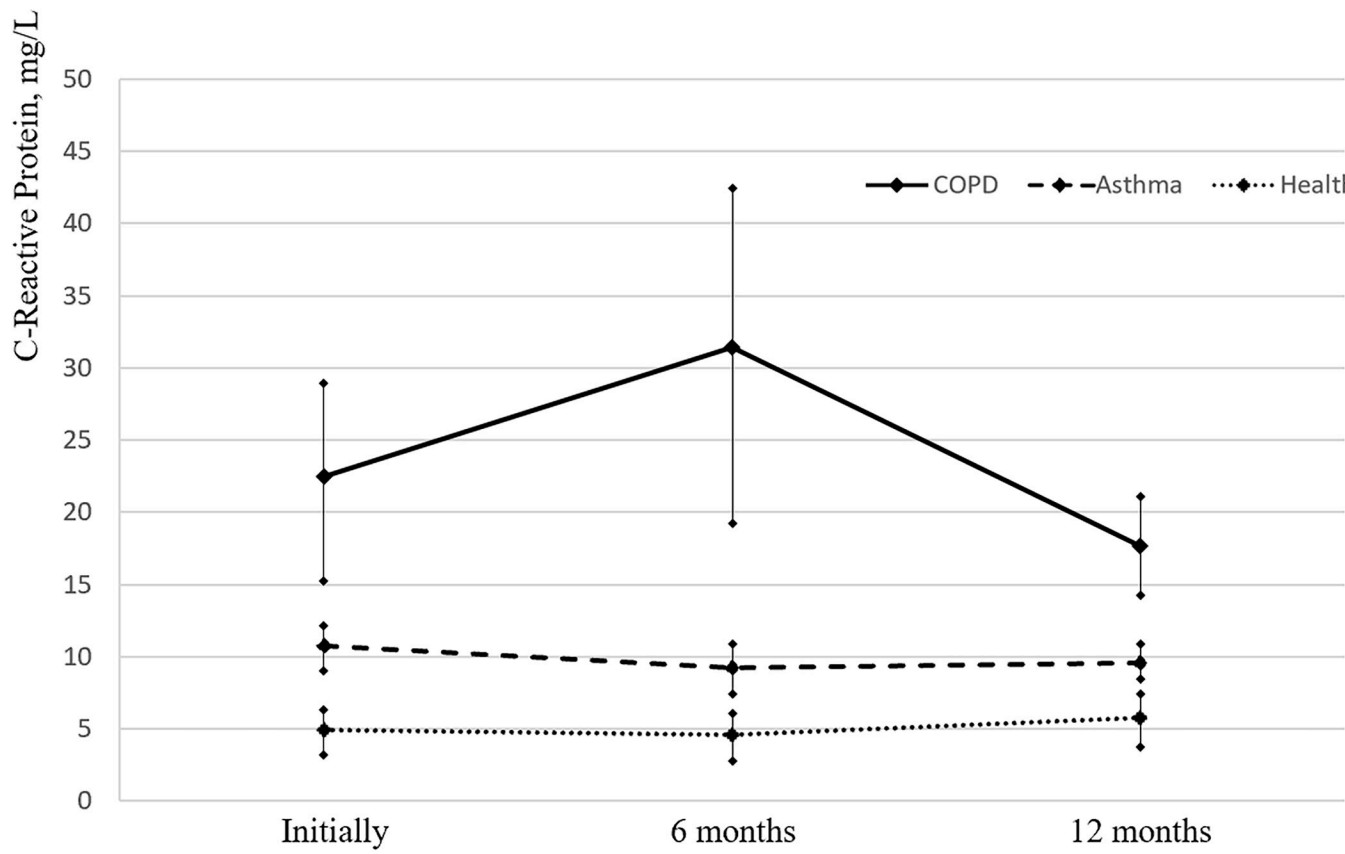

**Fig 1. Evolution of CRP levels in asthma and COPD patients and healthy individuals before and after influenza vaccination (normal values less than 8 mg/l).**

By the way it is important to notice that throughout the entire observation period, IL2, IL-17, and IL-10 remained within the reference ranges in all study groups.

## Correlation analysis of clinical, immunological, and functional parameters in asthma and COPD patients before vaccination against influenza

**Asthma patients.** In asthma patients before the vaccination, the IL-6 level is directly correlated with the level of serum CRP (r = 0.68; p <0.05. IL-6 also has a direct moderate correlation with the duration of exacerbations of the underlying disease (r = 0.54; p <0.05) and the number of systemic corticosteroids courses during exacerbations (r = 0.58; p < 0.05). A similar direct moderate correlation dependence was found between pro-inflammatory IL-2 and the frequency of systemic corticosteroids prescription (r = 0.68; p < 0.05). Moderate inverse correlations between the IL-17 level and FEV1 were revealed (r = -0.65; p <0.05).

**Table 5. The IL-2 dynamics in asthma and COPD patients vaccinated against influenza and healthy volunteers (reference value: Less than 10 pg/mL).**

| Group | n | Baseline | After 6 months | After 12 months | p |
|-------|---|----------|----------------|-----------------|---|
| | | M ± m | M ± m | M ± m | |
| COPD | 16 | 5.49 ± 4.19 | 1.41 ± 0.61 | 0.6 ± 0.27 | 0.15 |
| Asthma | 18 | 0.58 ± 0.31 | 0.35 ± 0.15 | 0.15 ± 0.1 | 0.37 |
| Healthy | 13 | 1.11 ± 0.47 | 0.55 ± 0.33 | 0.2 ± 0.11 | 0.49 |

Note: the significance of differences between the three evaluations in each group, the Friedman test was used

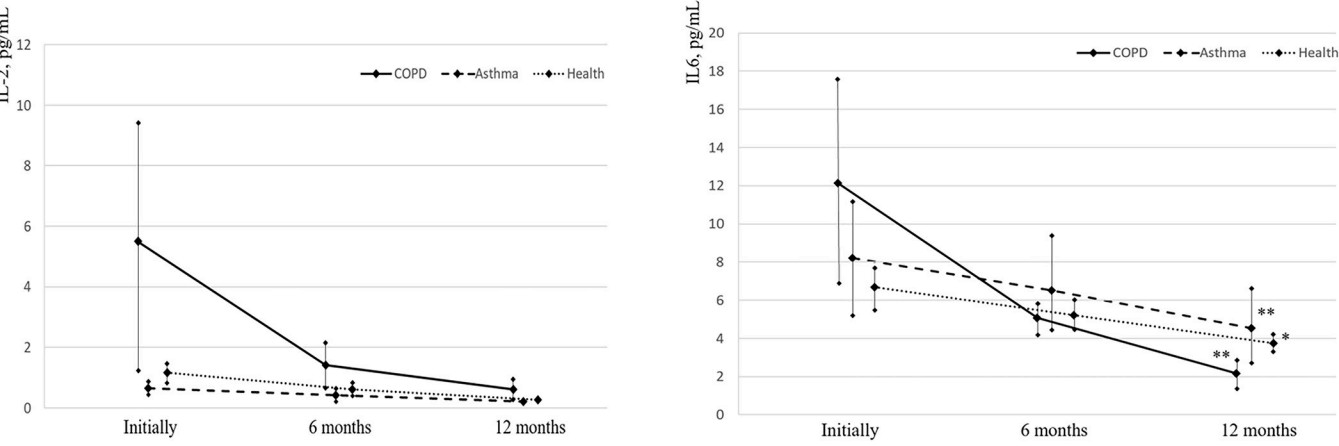

**Fig 2. Evolution of IL-2 and IL-6 levels in asthma and COPD patients and healthy individuals before and after influenza vaccination (normal values less than 10 pg/mL); * p< 0.05 ** p<0.005—significance of differences relative to the baseline values.**

**Table 6. Dynamics of IL-6 level in asthma and COPD patients vaccinated against influenza and healthy individuals (reference value: Less than 10 pg/mL).**

| Group | n | Quant./ | Baseline | After 6 months | | After 12 months | | p |
|---|---|---|---|---|---|---|---|---|
| | | % [1] | M ± m | M ± m | p | M ± m | p | |
| COPD | 18 | Quant. | 12.13 ± 5.69# | 5.06 ± 0.93 | 0.53 | 2.16 ± 0.77 | 0.004 | 0.025 |
| | | % | 14(78%) | 0 | 0.004 | 0 | 0.004 | <0.001 |
| Asthma | 18 | Quant. | 8.3 ± 3.08 | 6.61 ± 2.43 | 0.17 | 4.64 ± 2 | 0.003 | 0.009 |
| | | % | 7(39%) | 3(17%) | 0.15 | 1(6%) | 0.04 | 0.03 |
| Healthy | 12 | Quant. | 6.67 ± 1.16 | 5.22 ± 0.82 | 0.24 | 3.75 ± 0.71 | 0.018 | 0.034 |
| | | % | 0 | 0 | - | 0 | - | - |

Note

[1] The proportion of patients with laboratory parameters upper the reference values

#The parameter exceeds the upper limit of the norm (less than 10 pg/mL)

p: between the three evaluations in each group. For quantitative variables, the Friedman test with Dunn's multiple comparison was used, for binary variable, a generalized linear model was used (mixed effects logistic regression).

**Table 7. Dynamics of the IL-17 level in athma and COPD patients vaccinated against influenza and healthy volunteers (reference value: Less than 5 pg/mL).**

| Group | n | Baseline | After 6 months | After 12 months | p |
|---|---|---|---|---|---|
| | | M ± m | M ± m | M ± m | |
| COPD | 15 | 0.48 ± 0.14 | 0.33 ± 0.14 | 0.46 ± 0.32 | 0.65 |
| Asthma | 18 | 0.98 ± 0.2 | 1.02 ± 0.29 | 0.42 ± 0.28 | 0.11 |
| Healthy | 11 | 0.04 ± 0.04 | 0.12 ± 0.06 | 0.09 ± 0.07 | 0.54 |

Note: the significance of differences between the three evaluations in each group, the Friedman test was used

There was a direct strong correlation between the number of exacerbations and their duration (r = 0.91; p <0.05), the number of hospitalizations for exacerbations (r = 0.74; p <0.05), a direct moderate correlation with the number of exacerbations of the underlying disease with the frequency of outpatient visits (r = 0.39; p <0.05), the duration of hospitalizations (r = 0.68; p <0.05), the number of courses of antibiotic therapy (r = 0.42; p <0.05), and the number of SGCS courses (r = 0.54; p <0.05).

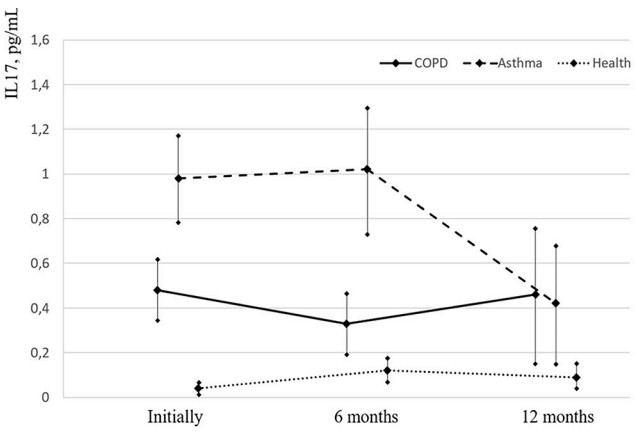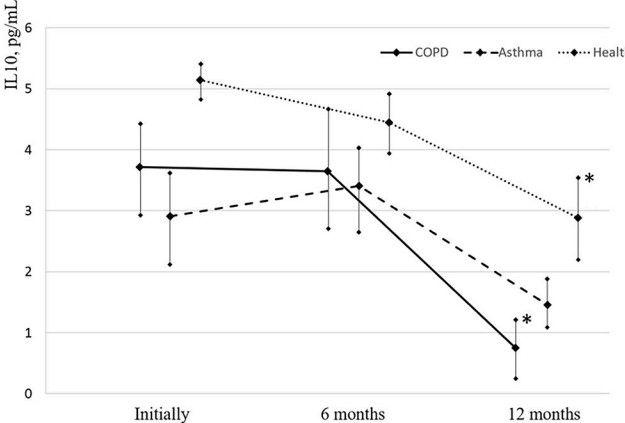

**Fig 3. Evolution of IL-17 (normal values less than 5 pg/mL) and IL-10 (normal values less than 20 pg/mL) levels in asthma and COPD patients and healthy individuals before and after influenza vaccination; * p< 0.05—significance of differences relative to the baseline values.**

**COPD patients.** In COPD patients before the vaccination against influenza, a strong inverse correlation between the pro-inflammatory cytokine IL-2 and the frequency of outpatient visits was revealed (r = -0.81; p<0.05).

A direct strong correlation was observed between the clinical parameters: the frequency of COPD exacerbations with the duration of exacerbations (r = 0.85; p <0.05), the frequency of outpatient visits (r = 0.73; p <0.05), and the frequency of hospitalizations (r = 0.78; p <0.05). The serum CRP level was directly correlated with the rate of FEV1 (r = 0.51; p <0.05). The frequency of hospitalizations had a direct moderate correlation with the frequency of prescription of antibacterial chemotherapy (r = 0.66; p <0.05).

## Discussion

The Russian trivalent polymer-subunit vaccine with a reduced (3-fold) content of epidemically relevant strains of influenza virus subtypes A (H1N1, H3N2) and type B (total 15 μg HA) and azoximer bromide (500 μg), was used in asthma and COPD patients. The vaccine has been used for more than 25 years within the framework of the National Immunization Schedule in children from 6 months of age without age restrictions, patients with various pathologies and pregnant [27–29]. Asthma and COPD patients were also vaccinated with this drug product, but it was not previously studied whether vaccination is accompanied by a change in the content of inflammatory mediators in the post-vaccination period and the considered immunological parameters.

During the follow-up period of asthma and COPD patients vaccinated against influenza, there were no unusual events that would be accompanied by a deterioration in the condition

**Table 8. Dynamics of the IL-10 level in asthma and COPD patients vaccinated against influenza and healthy volunteers (reference value: Less than 20 pg/mL).**

| Group | n | Baseline | After 6 months | | After 12 months | | p |
|---|---|---|---|---|---|---|---|
| | | M ± m | M ± m | p | M ± m | p | |
| COPD | 18 | 3.72 ± 0.73 | 3.65 ± 1.0 | 0.46 | 0.75 ± 0.51 | 0.013 | 0.048 |
| Asthma | 18 | 2.91 ± 0.77 | 3.4 ± 0.66 | 0.74 | 1.48 ± 0.44 | 0.084 | 0.26 |
| Healthy | 15 | 5.15 ± 0.33 | 4.46 ± 0.5 | 0.25 | 2.91 ± 0.73 | 0.006 | 0.013 |

Note: the significance of differences between the three evaluations in each group, the Friedman test with Dunn's multiple comparison was used

of patients, both in the early stage after the vaccination and the long-term period (12 months of the follow-up). On the contrary, within a year after the vaccination against influenza, was indicated a favorable effect on the clinical course of the disease with a 1.9-2-fold decrease in the frequency of BF exacerbations in asthma and COPD patients, respectively. In the same period, duration of exacerbations decreased by 2.2–2.5 times, frequency of hospitalizations decreased by 2.0–2.5 times, duration of hospitalizations decreased by 2.3–3 times, and the need for outpatient care decreased by 1.6 times in asthma patients and 20% (p >0.05) in the COPD patients. The obtained results are consistent with the data of other researchers who studied the vaccine efficacy against influenza in patients with bronchopulmonary pathology [30–33]. That is why it is completely justified according to the publication to give such relevant suggestions and recommendations of patient quality of life improvement not only for health care workers, but for health care managers.

Unlike asthma patients, the group of COPD patients within 12 months after the vaccination showed a 3.6-fold decrease in the number of courses of antibacterial drugs, which may indicate a decrease in infection-dependent COPD exacerbations. Similar results were obtained by Poole P.J. during vaccination of COPD patients against influenza [34] and Sumitani during combined vaccination against influenza and pneumococcal infection [35]. A.D. Protasov used combined vaccination against influenza, hemophilic and pneumococcal infections in COPD patients and obtained even more significant clinical effect: the number of exacerbations per year decreased by 3.7 times, and the number of courses of antimicrobial chemotherapy decreased by 4.3 times [36, 37].

Having been traditionally utilized as a marker of infection and cardiovascular events, there is now growing evidence that CRP plays important role in inflammatory processes and host responses to infection including the complement pathway, apoptosis, phagocytosis, nitric oxide (NO) release, and the production of cytokines, particularly interleukin-6 and tumor necrosis factor-$\alpha$ [38]. In unvaccinated patients the CRP level has a direct moderate correlation with the duration of exacerbations of the underlying disease with FEV1, which indicates that a reduction of inflammation in the airways leads to a decrease in acute-phase proteins, therefore, to a decrease in the duration of exacerbation.

The analysis of the CRP level as a marker of systemic inflammation and a predictor of prognosis in COPD within a year after the vaccination against influenza showed that in patients with BF the level of CRP exceeds the permissible values (especially in patients with COPD) as compared to the healthy vaccinated participants. No statistically significant changes in CRP parameters, after 6 and 12 months after the vaccination, in comparison with the baseline values, were revealed in any of the study groups, which may indicate a stable persistence of the chronic inflammatory process in the vaccinated BF patients.

IL-2, IL-6, and IL-17 are systemic pro-inflammatory cytokines, and IL-10 is an anti-inflammatory cytokine, the concentration of which increases in acute inflammatory processes and exacerbations of chronic diseases caused by a viral or bacterial infection.

IL-2 is a cytokine that regulates specific immune responses, binds to certain receptors on target cells, stimulates the growth, differentiation, and proliferation of T- and B-lymphocytes, monocytes, and macrophages. It stimulates the cytolytic activity of natural killer cells and cytotoxic T-lymphocytes and enhances the immune response (incl. antibacterial, antiviral response). The normal IL-2 concentration is not more than 10 pg/mL. IL-6 is a pro-inflammatory cytokine that is involved in the non-specific protection of the body against bacterial and viral infections. The IL-6 level depends on the severity of the inflammatory reaction, increases in the acute period of the disease followed by its decrease to normal. Its levels are increased in COPD patients and may be associated with COPD progression. In patients with frequent exacerbations, the levels of inflammatory markers in the remission phase are higher. The normal

concentration of IL-6 in the blood serum is 0–10 pg/mL, an average of 2.0 pg/mL. IL-17 enhances the secretion and action of pro-inflammatory cytokines. The study of IL-17 in BA showed its significant increase. Histamine and serotonin increase the production of IL-17. Studies have shown that animals lacking IL-17 are incapable of a rapid immune response and are more susceptible to infections, especially bacterial infections. At the same time, an excess of IL-17 can lead to the transition of the inflammatory process caused by the infection to the chronic phase. The normal concentration of IL-17 in the blood serum is less than 5 pg/mL.

IL-10 is an anti-inflammatory cytokine that regulates inflammatory reactions that develop during a specific immune response. IL-10 can suppress the function of monocytes, reduce the production of gamma-interferon, IL-1, and IL-8, and can stimulate IgE synthesis. As a result, it contributes to the development of the humoral component of the immune response, causing allergic reactivity of the body. Since IL-10 is an anti-inflammatory cytokine, there is no unambiguous opinion about how positive a reduction in its concentration during treatment is. Some studies consider a reduction in the concentration of anti-inflammatory IL (in particular, IL-10 in induced sputum in severe BA) as a defect of the anti-inflammatory response, which can lead to the uncontrolled synthesis of pro-inflammatory cytokines and the development of severe forms of the disease [38]. On the other hand, a reduction in the concentration of anti-inflammatory cytokines in response to a reduction in pro-inflammatory markers indicates a decrease in the systemic inflammatory response. The normal IL-10 concentration is less than 20 pg/mL. The study of cytokine levels in asthma patients and COPD patients vaccinated against influenza showed, that most parameters at the time of vaccination and during the study were within the proper values. A decrease in the level of the pro-inflammatory cytokine IL-6 and a tendency to a decrease in IL-2, IL-17 over time indicates a decrease in the systemic inflammatory process in vivo.

In asthma patients, the concentration of IL-17 was significantly higher than in COPD patients and healthy participants, that confirms the role of this cytokine in the development of asthma. The limitation of the study is that during the recruitment period of patients there was no assessment of asthma phenotype (Th2-low or Th2-high). Neutrophilic inflammation, in which IL-17 plays a leading role, is the most common reason for T2-low asthma. After 12 months post-vaccination in asthma patients, there was a tendency to a decrease in IL-17. In COPD patients, the level of IL-17 practically did not change during the observation period. In healthy participants, the content of IL-17 was lower than in COPD and asthma patients and did not exceed 0.12 ± 0.06 pg/mL.

We consider the decrease in anti-inflammatory IL-10, after the administration of the influenza vaccine as a response to the reduction in pro-inflammatory cytokines, since the regulation of the inflammatory response is carried out according to the principle of negative correlation. We observed a tendency towards a decrease in the level of pro-inflammatory cytokines IL-2, IL-6 in the first 6 months after the vaccination, which led to a decrease in the level of anti-inflammatory IL-10 in the next 6 months.

Previous studies assessing the *in vitro* mononuclear cell production of cytokines induced by influenza vaccines showed that all the investigated vaccines increased, to some extent, the levels of Th1/Th2/Th17/Th9/Th22 cytokines in cultures of mononuclear cells. Therefore, vaccination against influenza is accompanied by the activation of both humoral and cellular immunity [39]. The production of Th1 cytokines (IL-12, INF-g, IL-2, IL-6, IL-1β, and TNF-α) was more significantly induced by the vaccine containing an immune adjuvant, while the subunit vaccine had a lesser effect on the levels of these cytokines compared to the vaccine with an immune adjuvant or the split vaccine. These data support the assumption that a vaccine immune adjuvant is a more potent inducer of cellular immunity. The subunit vaccine more potently stimulated the production of IL-4, and the split vaccine had a greater effect on the

production of IL-5, both of which are secreted by Th2-cells. These findings could account for the positive changes in the clinical and functional parameters of asthma and COPD reported over the follow-up period [39, 40].

For a comprehensive assessment of the efficacy of vaccination, a correlation analysis of immunological, clinical, and functional parameters was carried out in BA and COPD patients against the background of vaccination against influenza. In both groups there were direct correlations of high and moderate strength between clinical parameters (frequency and duration of exacerbations, frequency and duration of hospitalizations, need for antibiotics and SGCS courses).

In patients before the vaccination, the level of IL-6 is directly correlated with the level of serum CRP, which is explained by the fact that the synthesis of CRP is carried out in hepatocytes and is regulated by interleukins IL-1 and IL-6. A direct correlation dependence of moderate strength was revealed betweenpro-inflammatory IL-2 and the frequency of systemic corticosteroids prescription. Moderate inverse correlations between the IL-17 level and FEV1 were revealed. This confirms the fact that excessive synthesis of IL-17 can lead to the chronicity of the inflammatory process and a decrease in FEV1.

The CRP level has a direct moderate correlation with the duration of exacerbations of the underlying disease with FEV1, which indicates that a reduction in inflammation in the airways leads to a decrease in acute-phase proteins, therefore, to a decrease in the duration of exacerbation.

A moderate inverse correlation was found between FEV1 and the number of exacerbations, duration of exacerbations, frequency of outpatient visits, number and duration of hospitalizations, and the number of SGCS courses. It can be assumed that an increase in functional parameters leads to a decrease in the need for medical care. A strong inverse correlation between the pro-inflammatory cytokine IL-2 and the frequency of outpatient visits was revealed.

After the administration of the influenza vaccine in COPD patients, a direct strong correlation between the level of IL-6 and the frequency of SGCS prescription during BF exacerbations was revealed. A decrease in the inflammatory response in the airways leads to a decrease in anti-inflammatory therapy.

It is important to emphasize that the immunoregulatory effect of the influenza vaccine on inflammatory mediators has confirmed its role in the activation of nonspecific immunity during the COVID-19 pandemic in numerous studies in which it was proven that seasonal influenza vaccination helped to reduce the susceptibility to SARS-CoV-2, severity of the disease and its outcomes [41–47].

Of note, the study sample is small, which makes it less representative, meaning that the study results are relevant for smaller homogeneous populations. To ensure homogeneity of the study population, we applied stricter inclusion and exclusion criteria compared to those used in larger studies (all study subjects were included within one epidemic period, they showed no signs of asthma or COPD exacerbation at the time of inclusion, etc.).

Thus, the analysis of immunological, clinical, and functional parameters in asthma and COPD patients confirms that vaccination against influenza in BF patients is effective and has a positive impact on the clinical, immunological, and functional parameters of the disease.

## Conclusion

The study evaluated the effect of a trivalent polymer-subunit vaccine with a reduced number of antigens associated with the azoximer bromide immunoadjuvant. The obtained results confirm the clinical vaccine efficacy for reducing the frequency of exacerbations, hospitalizations and increasing the remission period, as with other influenza vaccines. Vaccination may be accompanied by a decrease in the level of inflammatory mediators, which, in our opinion, is

an indirect effect, that is caused by a decrease in respiratory infections, the achievement of persistent remission of chronic bronchopulmonary pathology in the post-vaccination period leading to the normalization of certain parameters of the immune system. Namely, the preventive effect of the vaccination against influenza is reflected in the clinical course of asthma and COPD. It is anticipated that this study will become an opening to this problem investigation on a larger scale and will be taken into account by our colleagues.

## Supporting information

**S1 Fig. Figures for correlation of asthma.**
(DOCX)

**S1 Table. (Dataset with individual values separately for patients with bronchial asthma and COPD, respectively) for more detailed information can be found in supporting information files.**
(XLSX)

**S2 Table. (Dataset with individual values separately for patients with bronchial asthma and COPD, respectively) for more detailed information can be found in supporting information files.**
(XLSX)

## Author Contributions

**Conceptualization:** Mikhail Kostinov, Alexander Chuchalin, Nelli Akhmatova, Lyudmila Shogenova.

**Data curation:** Mikhail Kostinov, Anna Chebykina, Alexander Cherdantsev, Irina Solov'eva, Anna Shmit'ko.

**Formal analysis:** Anna Chebykina, Isabella Khrapunova, Alexander Cherdantsev, Irina Solov'eva, Valentina Polishchuk, Anna Shmit'ko.

**Investigation:** Mikhail Kostinov, Anna Chebykina, Nelli Akhmatova.

**Methodology:** Alexander Chuchalin, Nadezhda Kryukova, Anna Vlasenko.

**Project administration:** Mikhail Kostinov, Alexander Chuchalin.

**Software:** Nelli Akhmatova, Aristitsa Kostinova, Anna Vlasenko.

**Supervision:** Alexander Chuchalin.

**Validation:** Isabella Khrapunova, Alexander Cherdantsev, Valentina Polishchuk.

**Visualization:** Irina Solov'eva, Valentina Polishchuk, Aristitsa Kostinova, Anna Vlasenko, Marina Loktionova, Yvette Albahansa, Lyudmila Shogenova.

**Writing – original draft:** Mikhail Kostinov, Nadezhda Kryukova, Anna Vlasenko.

**Writing – review & editing:** Isabella Khrapunova, Alexander Cherdantsev, Marina Loktionova, Lyudmila Shogenova.

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
