## [Decision Letter · Decision Letter 0]

12 Jun 2024

PONE-D-24-13314Clinical status and cytokine profiles in patients with asthma or chronic obstructive pulmonary disease vaccinated against influenzaPLOS ONE

Dear Dr. Kostinov,

Thank you for submitting your manuscript to PLOS ONE. After careful consideration, we feel that it has merit but does not fully meet PLOS ONE’s publication criteria as it currently stands. Therefore, we invite you to submit a revised version of the manuscript that addresses the points raised during the review process.

**ACADEMIC EDITOR: **Abbreviations should be used judiciously as many are not widely recognized. The brand name of the influenza vaccine must be omitted from the submission, and conflicts of interest should be more explicitly stated for the relevant authors.

We look forward to receiving your revised manuscript.

Kind regards,

Bharat Bhushan Sharma, M.D.

Academic Editor

PLOS ONE

Journal Requirements:

**Additional Editor Comments:**

The editorial board and external reviewers have evaluated your submission. The manuscript may be reconsidered after appropriate revisions. Abbreviations should be used judiciously as many are not widely recognized. The brand name of the influenza vaccine must be omitted from the submission, and conflicts of interest should be more explicitly stated for the relevant authors.

Reviewers' comments:

Reviewer's Responses to Questions

**Comments to the Author**

1. Is the manuscript technically sound, and do the data support the conclusions?

Reviewer #1: Partly

Reviewer #2: Yes

2. Has the statistical analysis been performed appropriately and rigorously? 

Reviewer #1: I Don't Know

Reviewer #2: No

3. Have the authors made all data underlying the findings in their manuscript fully available?

Reviewer #1: No

Reviewer #2: Yes

4. Is the manuscript presented in an intelligible fashion and written in standard English?

Reviewer #1: Yes

Reviewer #2: Yes

5. Review Comments to the Author

Reviewer #1: This study investigates the lung function parameters and cytokine levels of influenza A and B virus vaccinated asthma and COPD patients compared to healthy vaccinated control subjects. Parameters were assessed 6 and 12 months after vaccination and compared to baseline levels obtained before vaccination. This study documented the course of pro-inflammatory cytokines IL-2, IL-6 and IL-17 as well as anti-inflammatory cytokine IL-10 in blood and correlated the results with patient hospitalization, medication and lung function. The authors confirm in their small subject study an improvement of asthma and COPD patients after Influenza virus vaccination that has been clinically observed although not documented in a publication. The publication could give relevant suggestions of patient treatment and quality of life improvement if these topics get included into the discussion part of the manuscript.

Asthma and COPD patients can have exacerbation events from bacterial infections with non-typable Haemophilus influenzae. Please specify in the introduction that “influenza” in this study is referring to Orthomyxoviridae and specifically Influenza A and B viruses.

Line 69: Please explain the meaning of the abbreviation “ARI” at line 69 instead of the current position at line 78.

Line 106: The heading “Methods” should be bold like “Materials” at line 224.

Line 135-138: “Within the next 12 months investigator - a pulmonologist, followed up all vaccinated patients and clinically assessed asthma and COPD course with repeated blood sampling and immunological studies in 6 and 12 months.” Can the authors provide more details on this procedure? Is it correct that only one pulmonologist performed all check-ups of all patients in the three different study centers mentioned in line 119-123?

Line 192: “The groups were comparable in terms of age, sex and a history of the disease (p >0.05).” It is suggested to rephrase this sentence. In Table 1a the distribution of male and female COPD patients is 95% and 5%, respectively. Also, healthy control were 20-10 years younger than asthma and COPD patients. Comparability is only given in regard of lung function parameters.

Line 216: “The vaccine forms a high-level specific immunity against influenza.” It is suggested to change the sentence to: “The vaccine forms a high-level specific immunity against the influenza strains mentioned above.”

Line 230-236 should be merged with the paragraph starting line 126 to line 141. The information of line 230-236 would benefit the paragraph starting line 126 as it describes the study design.

Line 263: The +/- symbol between Mean and SD is lost. Please insert the symbol anew.

Line 313: “Table 3, Figure 1, shows that the level of CRP exceeded the conditional norm (up to 8 mg/L) in all study periods in asthma and COPD patients.” It is suggested to change this sentence to “The level of CRP exceeded the conditional norm (up to 8 mg/L) in all study periods in asthma and COPD patients. (Table 3, Figure 1)”

Can the authors explain if the study subjects encountered viral/bacterial/fungi infections during the study course and how it was considered in the cytokine parameters assessed? Can the authors exclude that changes in cytokine levels occur due to infections not only in the lung but elsewhere in the body? Have the authors recorded other viral/bacterial/fungi infections in their patient’s history over the course of study and was it used as an inclusion/exclusion criteria?

Line 389-419: “Correlation analysis of clinical, immunological, and functional parameters in asthma and COPD patients before vaccination against influenza” The authors are advised to include the graphs of the correlations in the supplemental material of the paper.

Line 469: Please correct the typing error of “vaules” to “values”.

Line 515: Can the authors elaborate on how IL-17 contributes to the asthma phenotypes and its role in allergic inflammation? The discussion should include IL-17 and its actions on neutrophil-mediated immunity, which has not been addressed by the authors in the current manuscript.

Line 534: Do the authors with “T1 cytokines” mean “TH1 cytokines”?

Line 537-543: The authors are asked to discuss their hypothesis of the effect on IL-4 and IL-5 in more detail as these cytokines were not measured in the patient populations. Please support the paragraph by literature.

Line 588-591: The conclusion is very general and does not reflect the main findings of the study. And, vaccine development was not mentioned in the manuscript before. The authors should either revise their conclusion or address vaccine development in the introduction and discussion.

Can the authors give an outlook on how the influenza A and B virus vaccination can improve the quality of life of asthma and COPD patients in regard to lung function gain, ameliorated pulmonary infections and reduced exacerbation events?

Can the authors comment on vaccination strategies for asthma and COPD patients, especially when having a history of exacerbations?

Reviewer #2: The authors present a well-structured study to investigate the influence of influenza vaccination on the clinical course and inflammatory markers in asthma and COPD patients. The descriptions of the general experimental design and results are clearly communicated. The findings are relatively reliable, as the authors acknowledge the limitations due to the smaller sample size and provide literature evidence to support the major findings.

However, I have a few comments relevant to the study:

1. The statistics described in the paper do not mention multiple testing correction (MTC). The study is prone to Type I error due to the complexity of its experimental design. It would be beneficial to apply MTC, at least for Tables 1 and 2, since the number of hypotheses is large and there are some borderline significant p-values. While the significance might not greatly change for those with extremely low p-values, it might not be the case for the borderline ones.

2. The claim of 100% power might be extremely misleading and needs paraphrasing. Although the observed effect size can be very high, it is almost impossible in statistics to have a power of 100%, no matter how high the results appear. In fact, the unrealistically high calculated power might indicate a limitation due to the smaller sample size, rather than evidence to justify the sample size issue.

3. The ± symbol is not correctly shown in line 263, while the same symbol is correctly displayed elsewhere.

6. PLOS authors have the option to publish the peer review history of their article (what does this mean?). If published, this will include your full peer review and any attached files.

Reviewer #1: No

Reviewer #2: No

---

## [Author Response · Author response to Decision Letter 0]

12 Aug 2024

Dear Academic Editor! Thank you very much for your valuable comments and taking the time to read and study this article from the bottom of my heart!

• Abbreviations should be used judiciously as many are not widely recognized. The brand name of the influenza vaccine must be omitted from the submission, and conflicts of interest should be more explicitly stated for the relevant authors.

We tried to decipher all the abbreviations that appear in the text of article to make it easier to read and understand. The brand name of the influenza vaccine was omitted from the submission. The “conflict of interest” statement was changed.

Reviewer 1 - Thanks a lot for your time and attention paid for our manuscript! Undoubtedly, all your valuable comments are taken into account! 

• Asthma and COPD patients can have exacerbation events from bacterial infections with non-typable Haemophilus influenzae. Please specify in the introduction that “influenza” in this study is referring to Orthomyxoviridae and specifically Influenza A and B viruses.

According to your commentaries changes have been made in the Introduction section.

We changed all your mentioned comments on the text! We are very grateful for them! A correction of the original text of the manuscript was carried out thanks to your valuable commentaries:

• Line 69: Please explain the meaning of the abbreviation “ARI” at line 69 instead of the current position at line 78.

Line 106: The heading “Methods” should be bold like “Materials” at line 224.

Line 135-138: “Within the next 12 months investigator - a pulmonologist, followed up all vaccinated patients and clinically assessed asthma and COPD course with repeated blood sampling and immunological studies in 6 and 12 months.” Can the authors provide more details on this procedure? Is it correct that only one pulmonologist performed all check-ups of all patients in the three different study centers mentioned in line 119-123?

Line 192: “The groups were comparable in terms of age, sex and a history of the disease (p >0.05).” It is suggested to rephrase this sentence. In Table 1a the distribution of male and female COPD patients is 95% and 5%, respectively. Also, healthy control were 20-10 years younger than asthma and COPD patients. Comparability is only given in regard of lung function parameters.

Line 216: “The vaccine forms a high-level specific immunity against influenza.” It is suggested to change the sentence to: “The vaccine forms a high-level specific immunity against the influenza strains mentioned above.”

Line 230-236 should be merged with the paragraph starting line 126 to line 141. The information of line 230-236 would benefit the paragraph starting line 126 as it describes the study design.

Line 263: The +/- symbol between Mean and SD is lost. Please insert the symbol anew.

Line 313: “Table 3, Figure 1, shows that the level of CRP exceeded the conditional norm (up to 8 mg/L) in all study periods in asthma and COPD patients.” It is suggested to change this sentence to “The level of CRP exceeded the conditional norm (up to 8 mg/L) in all study periods in asthma and COPD patients. (Table 3, Figure 1)”

Line 469: Please correct the typing error of “vaules” to “values”. 

Line 534: Do the authors with “T1 cytokines” mean “TH1 cytokines”?

Line 588-591: The conclusion is very general and does not reflect the main findings of the study. And, vaccine development was not mentioned in the manuscript before. The authors should either revise their conclusion or address vaccine development in the introduction and discussion.

• Can the authors explain if the study subjects encountered viral/bacterial/fungi infections during the study course and how it was considered in the cytokine parameters assessed? Can the authors exclude that changes in cytokine levels occur due to infections not only in the lung but elsewhere in the body? Have the authors recorded other viral/bacterial/fungi infections in their patient’s history over the course of study and was it used as an inclusion/exclusion criteria?

Thank you for this important and clarifying question! This study did not assess the influence separately viral/bacterial/fungi infections on the cytokine parameters. During the observation period of the study, no viral, bacterial, or fungal infections requiring hospitalization, as well as additional diagnostics or consultation of infectious disease specialist were not conducted (during patients’ interview by the monitoring pulmonologists at noted above points of the study). In this study, systemic viral, bacterial and fungal infections were exclusion criteria both during the recruitment period and if they have occurred. However, we have currently accumulated material and are at the stage of processing on the influence of viral and bacterial infections on the cytokine profile of immunized patients with asthma and COPD after influenza vaccination in another study. 

• Line 389-419: “Correlation analysis of clinical, immunological, and functional parameters in asthma and COPD patients before vaccination against influenza” The authors are advised to include the graphs of the correlations in the supplemental material of the paper.

Thanks to your remark we have changed and included in Supplement 1!

• Line 515: Can the authors elaborate on how IL-17 contributes to the asthma phenotypes and its role in allergic inflammation? The discussion should include IL-17 and its actions on neutrophil-mediated immunity, which has not been addressed by the authors in the current manuscript.

The text was added: “The limitation of the study is that during the recruitment period of patients there was no assessment of asthma phenotype (Th2-low or Th2-high). Neutrophilic inflammation, in which IL-17 plays a leading role, is the most common reason for T2-low asthma.”

• Line 537-543: The authors are asked to discuss their hypothesis of the effect on IL-4 and IL-5 in more detail as these cytokines were not measured in the patient populations. Please support the paragraph by literature.

References have been added.

• Can the authors comment on vaccination strategies for asthma and COPD patients, especially when having a history of exacerbations?

Currently, vaccine prevention of pneumococcal and respiratory syncytial virus (RSV) infection, influenza and whooping cough is recommended for patients with asthma in accordance with GINA 2024. So we can say about global vaccination strategy in patients with asthma. In GINA 2024 it is written “Encourage children, adults and the elderly with asthma to follow their local immunization schedule, including for pneumococcal, pertussis, influenza, RSV and COVID-19 vaccinations.”

The same is for patients with COPD. In GOLD 2023 it is written “Influenza vaccination decreases the incidence of lower respiratory tract infections.”

The recommendations on vaccine prevention are written in guidelines of ATS, AAACI and other countries’ position papers. In Russia we have also Federal clinical recommendations, laws and documents concerning vaccine prophylaxis of influenza and other respiratory infections (in Russian) in immunocompromised patients, especially with bronchopulmonary diseases. But even nowadays in the age of personalized medicine, there is no specific recommendations regarding the type of vaccines that would be preferred in patients with different pathologies. And an individual approach to each patient is essential to achieve the most optimal, best effect.

Reviewer 2 – Many thanks for your in-depth analysis of our article and investigation! It is an honor for us!

• The statistics described in the paper do not mention multiple testing correction (MTC). The study is prone to Type I error due to the complexity of its experimental design. It would be beneficial to apply MTC, at least for Tables 1 and 2, since the number of hypotheses is large and there are some borderline significant p-values. While the significance might not greatly change for those with extremely low p-values, it might not be the case for the borderline ones.

The Benjamini–Hochberg procedure was used (Table 1, Table 2) to control for the false discovery rate at a 5% level. In the Table 1 nothing have changed after introducing the amendment, but in the Table 2 the change in VC, % became statistically insignificant!

• The claim of 100% power might be extremely misleading and needs paraphrasing. Although the observed effect size can be very high, it is almost impossible in statistics to have a power of 100%, no matter how high the results appear. In fact, the unrealistically high calculated power might indicate a limitation due to the smaller sample size, rather than evidence to justify the sample size issue.

Thanks to your remark we have paraphrase this sentence and will be more attentive to the meaningful details! This phrase was removed and instead inserted into the materials and methods: “All available data on the database were used to maximize the power and generalizability of the results [Cuschieri, Sarah. "The STROBE guidelines." Saudi journal of anaesthesia 13.Suppl 1 (2019): S31-S34].” 

• The ± symbol is not correctly shown in line 263, while the same symbol is correctly displayed elsewhere.

Done! Thanks a lot for your attentive revision!

---

## [Decision Letter · Decision Letter 1]

25 Sep 2024

PONE-D-24-13314R1Clinical status and cytokine profiles in patients with asthma or chronic obstructive pulmonary disease vaccinated against influenzaPLOS ONE

Dear Dr. Kostinov,

Thank you for submitting your manuscript to PLOS ONE. After careful consideration, we feel that it has merit but does not fully meet PLOS ONE’s publication criteria as it currently stands. Therefore, we invite you to submit a revised version of the manuscript that addresses the points raised during the review process.

**ACADEMIC EDITOR: **There remain some methodological concerns that warrant attention. Analyzing binary response and explanatory variables is more effectively conducted through logistic regression, which establishes a curvilinear model relationship, as opposed to a linear probability model. This approach will enhance the manuscript's utility.

We look forward to receiving your revised manuscript.

Kind regards,

Bharat Bhushan Sharma, M.D.

Academic Editor

PLOS ONE

Journal Requirements:

Additional Editor Comments:

There remain some methodological concerns that warrant attention. Analyzing binary response and explanatory variables is more effectively conducted through logistic regression, which establishes a curvilinear model relationship, as opposed to a linear probability model. This approach will enhance the manuscript's utility.

Reviewers' comments:

Reviewer's Responses to Questions

**Comments to the Author**

1. If the authors have adequately addressed your comments raised in a previous round of review and you feel that this manuscript is now acceptable for publication, you may indicate that here to bypass the “Comments to the Author” section, enter your conflict of interest statement in the “Confidential to Editor” section, and submit your "Accept" recommendation.

Reviewer #2: All comments have been addressed

2. Is the manuscript technically sound, and do the data support the conclusions?

Reviewer #2: Yes

3. Has the statistical analysis been performed appropriately and rigorously? 

Reviewer #2: I Don't Know

4. Have the authors made all data underlying the findings in their manuscript fully available?

Reviewer #2: Yes

5. Is the manuscript presented in an intelligible fashion and written in standard English?

Reviewer #2: Yes

6. Review Comments to the Author

Reviewer #2: I am generally satisfied with the responses and updates, and I appreciate the amendments made. However, I would like to remind you to mention the use of multiple corrections testing, update and report the adjusted p-values for both Table 1 and Table 2, and highlight the significant variables based on the adjusted p-values rather than the original p-values. Otherwise, the purpose of performing the Benjamini–Hochberg procedure would be undermined, and your manuscript might contradict your responses to the reviewers (for example, VC, % is still highlighted as significant). This is the only point I suggest modifying further in this revised version.

7. PLOS authors have the option to publish the peer review history of their article (what does this mean?). If published, this will include your full peer review and any attached files.

Reviewer #2: No

---

## [Author Response · Author response to Decision Letter 1]

19 Oct 2024

1. There remain some methodological concerns that warrant attention. Analyzing binary response and explanatory variables is more effectively conducted through logistic regression, which establishes a curvilinear model relationship, as opposed to a linear probability model. This approach will enhance the manuscript's utility.

- Categorization of spirometry parameters relative to reference values is not informative, since all parameters in 100% of patients are below LLN (lower limit of normal). It is worth noting that Table 2 shows the values as a percentage of LLN, which allows us to evaluate the dynamics of parameters relative to reference values.

A similar situation is also observed with IL2, IL-17 and IL-10 parameters. In all 100% of patients, these parameters did not exceed the upper limit of the reference range throughout the study period. In accordance with the editor's recommendations, this was added to the text of the article.

CRP and IL-6 parameters were categorized relative to the reference range and the dynamics of the proportion of patients with parameter values above the reference were assessed (Table 3, Table 5).

2. I am generally satisfied with the responses and updates, and I appreciate the amendments made. However, I would like to remind you to mention the use of multiple corrections testing, update and report the adjusted p-values for both Table 1 and Table 2, and highlight the significant variables based on the adjusted p-values rather than the original p-values. Otherwise, the purpose of performing the Benjamini–Hochberg procedure would be undermined, and your manuscript might contradict your responses to the reviewers (for example, VC, % is still highlighted as significant). This is the only point I suggest modifying further in this revised version.

- Thanks a lot for the remark! Done!

---

## [Editor Report · Decision Letter 2]

28 Oct 2024

Clinical status and cytokine profiles in patients with asthma or chronic obstructive pulmonary disease vaccinated against influenza

PONE-D-24-13314R2

Dear Dr. Kostinov,

We’re pleased to inform you that your manuscript has been judged scientifically suitable for publication and will be formally accepted for publication once it meets all outstanding technical requirements.

Kind regards,

Bharat Bhushan Sharma, M.D.

Academic Editor

PLOS ONE

Additional Editor Comments (optional):

The authors have addressed all the questions posed during the peer review process.
---

## [Editor Report · Acceptance letter]

19 Nov 2024

PONE-D-24-13314R2 

PLOS ONE

Dear Dr. Kostinov, 

I'm pleased to inform you that your manuscript has been deemed suitable for publication in PLOS ONE. Congratulations! Your manuscript is now being handed over to our production team.

Kind regards, 

on behalf of

Professor Bharat Bhushan Sharma 

Academic Editor

PLOS ONE